# The Molecular Effects of a High Fat Diet on Endometrial Tumour Biology

**DOI:** 10.3390/life10090188

**Published:** 2020-09-10

**Authors:** Michael Wilkinson, Piriyah Sinclair, Ludmilla Dellatorre-Teixeira, Patrick Swan, Eoin Brennan, Bruce Moran, Dirk Wedekind, Paul Downey, Kieran Sheahan, Emer Conroy, William M. Gallagher, Neil Docherty, Carel le Roux, Donal J. Brennan

**Affiliations:** 1Department of Gynaecological Oncology, UCD School of Medicine, Mater Misericordiae Universtity Hospital, Eccles Street, Dublin 7, D07 AX57 Dublin, Ireland; michael.wilkinson@ucdconnect.ie; 2UCD Diabetes Complications Research Centre, UCD Conway Institute, University College Dublin, D14 NN96 Dublin, Ireland; piriyah.sinclair@ucdconnect.ie (P.S.); ludmilla.pessanha@ucd.ie (L.D.-T.); patrick.swan@ucdconnect.ie (P.S.); eoin.brennan@ucd.ie (E.B.); neil.docherty@ucd.ie (N.D.); 3Department of Pathology, St Vincent’s University Hospital, Elm Park, Dublin 4, D04 YN63 Dublin, Ireland; bruce.moran@ucd.ie (B.M.); ksheahan@svhg.ie (K.S.); 4Biomedical Facility, Hanover Medical School, 30625 Hanover, Germany; Wedekind.Dirk@mh-hannover.de; 5Department of Pathology, National Maternity Hospital, Holles Street, Dublin 2, D02 YH21 Dublin, Ireland; pdowney@nmh.ie; 6Cancer Biology and Therapeutic Laboratory, UCD School of Biomolecular and Biomedical Science Ireland, UCD Conway Institute, University College Dublin, D14 NN96 Dublin, Ireland; emer.conroy@ucd.ie (E.C.); William.Gallagher@ucd.ie (W.M.G.); 7Systems Biology Ireland, UCD School of Medicine, Belfield, Dublin 4, D14 NN96 Dublin, Ireland

**Keywords:** endometrial cancer, obesity, transcriptomics

## Abstract

We sought to validate the BDII/Han rat model as a model for diet-induced obesity in endometrial cancer (EC) and determine if transcriptomic changes induced by a high fat diet (HFD) in an EC rat model can be used to identify novel biomarkers in human EC. Nineteen BDII/Han rats were included. Group A (*n* = 7) were given ad lib access to a normal calorie, normal chow diet (NCD) while Group B (*n* = 12) were given ad lib access to a calorie rich HFD for 15 months. RNAseq was performed on endometrial tumours from both groups. The top-ranking differentially expressed genes (DEGs) were examined in the human EC using The Cancer Genome Atlas (TCGA) to assess if the BDII/Han rat model is an appropriate model for human obesity-induced carcinogenesis. Weight gain in HFD rats was double the weight gain of NCD rats (50 g vs. 25 g). The incidence of cancer was similar in both groups (4/7—57% vs. 4/12—33%; *p* = 0.37). All tumours were equivalent to a Stage 1A, Grade 2 human endometrioid carcinoma. A total of 368 DEGs were identified between the tumours in the HFD group compared to the NCD group. We identified two upstream regulators of the DEGs, *mir-33* and *Brd4*, and a pathway analysis identified downstream enrichment of the colorectal cancer metastasis and ovarian cancer metastasis pathways. Top-ranking DEGs included *Tex14, A2M, Hmgcs2, Adamts5, Pdk4, Crabp2, Capn12, Npw, Idi1* and *Gpt. A2M* expression was decreased in HFD tumours. Consistent with these findings, we found a significant negative correlation between *A2M* mRNA expression levels and BMI in the TCGA cohort (Spearman’s Rho = −0.263, *p* < 0.001). *A2M* expression was associated with improved overall survival (HR = 0.45, 95% CI 0.23–0.9, *p* = 0.024). *Crabp2* expression was increased in HFD tumours. In human EC, *CRABP2* expression was associated with reduced overall survival (HR = 3.554, 95% CI 1.875–6.753, *p* < 0.001). Diet-induced obesity can alter EC transcriptomic profiles. The BDII/Han rat model is a suitable model of diet-induced obesity in endometrial cancer and can be used to identify clinically relevant biomarkers in human EC.

## 1. Introduction

Endometrial cancer is the most common cancer of the female genital tract and the fourth commonest cancer in women worldwide. In 2016, in the United States of America, there were 60,050 new cases of endometrial cancer and over 10,000 deaths [1]. The lifetime risk of endometrial cancer in women in Europe is thought to be between 1.7 and 2%, equating to a lifetime risk of approximately 1/80 [2]. The incidence of endometrial cancer has increased by over 20% since 2008, owing in large part to the growing epidemic of obesity and diabetes, both of which are considerable risk factors for endometrial cancer [2].

Obesity is the greatest risk factor for endometrial cancer, and there is a growing need to understand its effect on transcriptomic alterations within the endometrium. Obesity is generally associated with low-grade [3,4], low mutational burden, microsatellite-stable, p53 wild-type adenocarcinomas of the endometrium, which usually develop on a background of complex atypical endometrial hyperplasia. Loss-of-function mutations in *PTEN* are found in the majority of low-grade endometrial tumours as well as in complex atypical hyperplasia, suggesting that they are an initiating event in tumorigenesis [5,6]. The most frequently mutated genes in low-grade endometrial cancer are *PTEN, PIK3CA, CTNNB1, ARID1A* and *PIK3R1* [7]. Gain-of-function mutations in *CTNNB1* exon 3 are particularly prevalent in young women with obesity, and a diagnosis of low-grade endometrioid cancer is associated with a poor prognosis [8].

Given the well-established interaction between obesity, metabolism and inflammation, immunocompetent animal models of endometrial cancer that replicate this obesity-related phenotype and genotype are necessary to improve our understanding of the underlying molecular mechanisms. The BDII/Han rat model is a spontaneous, immunocompetent model in which up to ninety percent of animals develop endometrial cancer during their lifetime (26–28 months) [9,10]. BDII/Han endometrial tumours express oestrogen and progesterone receptors [11,12], respond to progestogen and are histologically similar to obesity-related endometrial cancers in humans, which were previously referred to as Type 1 tumours [13]. Endometrial tumours from BDII/Han rats have many molecular similarities to obesity-related human endometrial cancer, including microsatellite stability [14], a lack of *Erbb2* mutations and reduced expression of *Pten*—albeit without evidence of loss-of-function mutations [15,16]. Unlike human low-grade tumours, p53 mutations have been reported in the BD/II Han rat model, occurring in up to 67% of cases, and this model is believed to reflect high-grade Type 1 endometrial cancers in humans [16].

As a result of the molecular and histological similarities between the BDII/Han rat model and human obesity-related endometrial carcinoma, we sought to examine the impact of diet-induced obesity on the incidence and molecular profiles of endometrial cancers arising in this model. The aims of our study were to assess if BDII/Han rats respond to a high fat diet and develop diet-induced obesity. We also sought to assess obesity-induced transcriptomic alterations in BD/II Han endometrial tumours and to ascertain if these alterations could be used to identify novel targets in human endometrial cancer.

## 2. Results

### 2.1. High Fat Diet-Induced Visceral Obesity in BD/II Han Rats

The average age of all the rats commencing this study was 106 days. The mean weight of the rats in the NCD group was 185.7 g (SD 8.0) and 177.2 g (SD 18.7) in the HFD group. The average age of both groups was similar at the study’s end: 451 days (range 446–459) in the NCD group and 442.8 days (range 429–454 days) in the HFD group.

This group remained on “ad lib” NCD and water for the study duration to an average age of 451.1 days (range 446–459). Weight gain in the HFD group was double the weight gain of the NCD group (50 g (SD 27.3) vs. 25 g (SD 7.5), *p* = 0.01). The average weight of the NCD group was 211 g (SD 6.4) vs. 227 g (SD 12.4), *p* = 0.01) in the HFD group (Figure 1A). This coincided with a subjective increase in abdominal and perivisceral fat in all animals in the HFD relative to animals in the NCD, indicating that diet-induced obesity can be achieved in this animal model. The increased volume in fat distribution was particularly noticeable in the omentum, peri-renal and peri-vesical areas across all rats in this group and was easily distinguishable on imaging (Figure 1B).

### 2.2. Diet-Induced Obesity Did Not Increase the Endometrial Cancer Burden in BDII/Han Rats

The overall incidence of endometrial cancer in this study was 42% (8 of 19 rats). The incidence of endometrial cancer in the NCD group was 57.1% (4 of 7 rats) and was 33.3% (4 of 12 rats) in the HFD (*p* = 0.37). There was no evidence of metastatic disease and all tumours were confined to the inner half of the myometrium. A blinded histological assessment did not identify any histological difference in the tumours from either group as all were Grade 2 endometroid adenocarcinomas with cellular pleomorphism and prominent nucleoli (Figure 1C,D).

### 2.3. Diet-Induced Obesity Caused Specific Transcriptomic Changes in BDII/Han Endometrial Tumours

A total of 26 RNA samples were isolated from tumours with 14 samples from the NCD tumours (*n* = 4) and 12 from the HFD group tumours (*n* = 4), respectively. Multiple samples were extracted from different tumours to control for heterogeneity (see Appendix A). The mean number of raw reads across all samples was 45 million (SD 2.2 million). Approximately 78.3% of the reads aligned to the *Rattus norvegicus* dataset (Rnor_6.0), with high quality scores (≥Q30). After normalization and scaling of gene expression differential gene expression analysis identified 383 statistically significant differentially expressed transcripts (FDR *p* < 0.05) in the HFD group relative to the NCD group, of which 368 were mapped to genes (Figure 2A). The top ten DEGs in the HFD group relative to the tumours in the NCD group are presented in Figure 2A.

We sought to investigate pathway enrichment within the significant DEGs from the HFD versus NCD groups comparison using an Ingenuity Pathway Analysis (IPA; Qiagen, MD, USA). There were 329 significantly enriched canonical pathways, of which 173 reached a threshold log-fold expression of 1.3, and with many reaching statistical significance (*p* < 0.001). The most significantly enriched pathways based on a Z-score > 2 or <−2 are presented in Figure 2B, which demonstrate the positively enriched pathways were p14/p19 ARF and the endocabinoid pathways, while the negatively enriched pathways were the colorectal cancer metastasis pathway (Z = −2.80) and ovarian cancer signalling pathways (Z = −2.54). The identification of p14/ARF as an enriched pathway is consistent with the fact that the expression levels of p14 increase in Grade 1 tumours compared to a normal endometrium and are then reduced in higher-grade tumours [17].

To further understand the impact of diet-induced obesity on the endometrial cancer transcriptome in BDII/Han rats, we used IPA to identify the putative upstream regulators of the 368 DEGs. We identified 13 enriched upstream regulators with a Z-score ≥ 2 or ≤ −2 (Figure 2C); however, only two reached statistical significance: *mir-33*, which was positively enriched in HFD group, and *Brd4*, which was negatively enriched in the HFD group (Figure 2C). Although little is known about *mir-33* in cancer, *mir-33* knockout mice are predisposed to obesity, metabolic dysfunction and insulin resistance, suggesting that *mir-33* activity may be a protective mechanism in rats exposed to an HFD [18].

*Brd4* is a member of the Bromodomain family of proteins, which play a key role in the epigenetic regulation of transcription and interacts with the oestrogen receptor in animal models of endometrial and breast cancer [19]. *Brd4* regulated the expression of five genes from our dataset: *Ednrb*, *Col5a2*, *Tlr8*, *Spon2* and *Pole4* (Figure 2D). Kaplan–Meier analysis demonstrated increased levels of *BRD4* were associated with decreased OS (*p* = 0.043) in the TCGA cohort (Figure 2E), which is consistent with previous findings that demonstrated that the *NSD3*–*CHD8*–*BRD4* pathway is amplified in approximately 9% of serous carcinomas, and this amplification leads to a reduced overall survival [20]. We also sought to assess BRD4 protein expression in endometrial cancer using the Human Protein Atlas; this demonstrated “strong” BRD4 protein expression in 9/11 endometrial cancers (Figure 2F). Univariate Cox regression analysis demonstrated that decreased mRNA expression of *BRD4* was associated with decreased OS (HR = 2.034, 95% CI 1.009–4.102, *p* = 0.047) in the TCGA endometrial cancer cohort however this did not remain significant in a multivariate models (Appendix A).

Taken together these data suggest that alterations induced by diet-induced obesity alters the transcriptome in a rat model of spontaneous endometrial cancer. We identified a well-established obesity regulator of lipid metabolism, *mir-33*, as an important upstream regulator in the HFD group, and we highlighted for the first time a potentially important role for *Brd4* in the regulation of genes altered by diet-induced obesity.

### 2.4. Obesity-Regulated Genes Can Be Used as Biomarkers in Human Endometrial Cancer

To assess the clinical relevance of our findings in the rat model, we examined the relationship between the 10 most differentially expressed genes identified in the rat model and the clinico-pathological variables in the TCGA dataset (Figure 3A). In human EC, expression levels of all 10 genes were distributed across all four human endometrial cancer molecular subtypes—POLE, microsatellite instable, copy number low and copy number high.

From these ten genes, Alpha-2 macroglobulin (*A2M*) and Cellular Retinoic Acid Binding Protein 2 (*CRABP2*) demonstrated the highest number of alterations in human EC (Figure 3A), and thus were chosen for further assessment. In our rat study, *A2M* expression was significantly reduced in endometrial tumours from the HFD compared to NCD group (Figure 2A). Consistent with these findings, we found a significant negative correlation between *A2M* mRNA expression levels and BMI in the TCGA cohort (Spearman’s Rho = −0.263, *p* < 0.001; Table 1). *A2M* alterations were identified in 13% of human endometrial cancers in the TCGA dataset (Figure 3A). All of the mutations identified were missense mutations of uncertain significance (Figure 3A). Regression tree analysis demonstrated the optimal threshold for *A2M* for survival analysis was a Log2TPM value of 7.303. Using this threshold, *A2M* mRNA expression was associated with low grade (*p* < 0.001) endometrioid tumours (*p* = 0.003; Table 1) and the Kaplan–Meier analysis demonstrated increased levels of *A2M* were associated with improved OS (*p* = 0.02; Figure 3B). Univariate Cox regression analysis demonstrated that increased expression of *A2M* was associated with improved OS (HR = 0.45, 95% CI 0.23–0.9, *p* = 0.024); however, this did not remain significant in a multivariate model (Table 2). We also assessed *A2M* protein expression in endometrial cancer using the Human Protein Atlas; this demonstrated “strong” *A2M* expression in 3/11 endometrial cancers (Figure 3C).

*Crabp2* expression was upregulated in endometrial tumours from rats fed an HFD. *CRABP2* alterations were identified in 13% of endometrial cancers in the TCGA dataset (Figure 3A). All of the mutations identified were missense mutations of uncertain significance. Regression tree analysis demonstrated the optimal threshold for *CRABP2* for survival analysis was a log2TPM value of 7.319. Using this threshold, *CRABP2* expression was associated with a high grade (*p* = 0.0001), serous histology (*p* = 0.001), advanced stage (*p* = 0.005) and the copy number high molecular subtype (*p* = 0.001; Table 1). The Kaplan–Meier analysis demonstrated that increased levels of *CRABP2* were associated with reduced OS (*p* ≤ 0.001; Figure 3D). Univariate Cox regression analysis demonstrated that increased expression of *CRABP2* was associated with reduced OS (HR = 3.554, 95% CI 1.875–6.753, *p* < 0.001; Table 1). Multivariate analysis demonstrated increased *CRABP2* expression was associated with a reduced OS after controlling for age, grade, histological subtype and FIGO stage (HR = 3.13, 95% CI 1.52–6.44, *p* = 0.002; Table 2). Assessment of protein expression in endometrial cancer using the Human Protein Atlas demonstrated CRABP2 expression in 3 of 11 endometrial cancers (Figure 3E).

### 2.5. Mutation and Expression of Tp53, Molecular Characterisation

To further define the rat model as being relevant to obesity-related human endometrial cancer, which generally is associated with functional p53, we assessed the *Tp53* variants using RNAseq in the rat model [21,22]. We found a highly prevalent *Tp53* frameshift variant, p.Pro320ThrfsTer10, extant in 37/65 samples, with 15 individual rats having at least one sample with the variant (Appendix A). A total of 15/18 (84%) samples from the NCD group were found with the variant compared with 22/47 (49%) in the HFD group. Because of this disparity, which could confound the groupings, we examined the role of this variant and other rare (essentially private) variants in relation to *Tp53* expression.

Samples with the p.Pro320ThrfsTer10 frameshift, located in the p53 tetramerisation protein domain, were not found to have significantly different expression of *Tp53* at the gene level or transcript level (5 expressed isoforms) compared to samples without (Wilcox tests). Samples with at least one of the 15 other variants found within the coding sequence of *Tp53* were also not found to have a significantly different expression at the gene-level, but two lowly expressed isoforms did show significant difference (see Appendix A). Of those 15 variants, 8 were found to have a direct human ortholog in the COSMIC database [23] (see Appendix A) and all occurred within the DNA-binding protein domain. No comparable human variant of the p.Pro320ThrfsTer10 frameshift was found in COSMIC.

Downstream genes regulated by *TP53* in humans were investigated using the MSigDB [24] HALLMARK_P53_PATHWAY, which contains 195 human genes with an appropriate rat ortholog. Using t-test with false discovery rate adjustment, we found no significant divergence in expression grouped by the mutant or wild-type p.Pro320ThrfsTer10 frameshift (*p* < 0.01). We found 2/195 genes with significantly divergent expression when grouping by presence of any other missense, frameshift or nonsense variant in *Tp53* vs. no variant. These were *Cdkn2a* and *Hexim1* (see Appendix A). We found only 8 total mutations in these genes, 7 of which occurred in *Cdkn2a* and none of which occurred in more than three samples. We do not expect that these mutations explain the observed divergent expression.

To further categorise the model, assuming it reflects a *Tp53* wild-type phenotype, we estimated microsatellite stability using the “preMSIm” R package [25]. This method uses the scaled expression of 15 human genes, of which 10 orthologs were extant in the rat model. We found 51/65 sample to be microsatellite instable (MSI) vs. microsatellite stable (MSS), with no significant difference between tumour and normal tissue (24, 27 MSI and 7, 7 MSS, respectively). Only two individual rats were MSS (i.e., no samples determined to be MSI), these contributing 7 total samples to the study, and one each in the different diet groups.

Finally, we investigated the overall variant levels. We annotated all variants and removed those with a low coverage (7 REF reads, 3 ALT reads) and denoted those that were fixed in the population. Of a total of 1,339,616 variants, 761,902 were denoted as fixed in the population, with 539,933 at low coverage with 46,833 overlap. A further 90 were removed as being multiallelic. Using the remaining 84,524 variants, 74% are private (i.e., only found in 1 sample) and 6% are universal (all samples; see Appendix A). We calculated mutational burden similarly to tumour mutation burden (TMB, [26]), taking all variants and dividing by the total region sequenced and variant-called (~854 megabases). Mean mutational burden was 10.7 (min 99.4, max 14.3), and was not significantly different in tissue (tumour vs. normal, Fisher Exact test). Taken together, these findings demonstrate that the BD/II Han EC model corresponds with a human microsatellite unstable model.

## 3. Discussion

Understanding obesity-related carcinogenesis is a research priority, as a clear understanding of the underlying molecular mechanisms will be a key foundation to any public health measures focused on weight loss as a preventative or therapeutic intervention. Here we present data on a well-established spontaneous immune-competent EC model. We demonstrate that BDII/Han rats are susceptible to diet-induced obesity, causing an increased volume of intrabdominal and perivisceral fat in all animals in the HFD group, suggesting that this model warrants further investigation. Unfortunately, diet-induced obesity did not result in a statistically significant increased tumour burden (57.1% vs. 33.3%, *p* = 0.37; Appendix A). This may be due to the small sample size; however, the overall incidence of cancer in the rats in this study was 49%, which is slightly lower than expected. [9]. A post-hoc power analysis showed that the study would require 116 rats to demonstrate a statistically significant difference in the endometrial cancer incidence in the NCD and HFD groups.

A second reason as to why diet-induced obesity did not increase the tumour burden in these rats is likely related to the underlying genetics of this model. In particular, we confirm previous findings [16] that p53 mutations are common in the BDII/Han tumours but these do not appear to be a key driver in this model. A pan-cancer analysis of human p53 recently showed a significantly decreased transcript and protein expression of p53 when truncating variants were present versus wild-type or missense variants [27]. The p.Pro320ThrfsTer10 frameshift found in our rat model occurs in the p53 tetramerisation domain and has no human equivalent. The majority of human p53 variants, as well as the other variants found in our data, occur in the p53 DNA-binding domain. We hypothesise that the prevalent frameshift found herein is not functional in truncating the protein based on the gene’s comparable expression to “true” wild-type samples. While two lowly-expressed isoforms are significantly different in the *Tp53* expression grouping on the non-p.Pro320ThrfsTer10 variants, the two major isoforms are unaffected. Further, the investigated *TP53* hallmark pathway was completely unaffected by p.Pro320ThrfsTer10 presence, and 2/195 genes were affected by the non-p.Pro320ThrfsTer10 variants, which could be due to non-p53-related effects.

In human EC, p53 mutations are associated with serous carcinoma and the copy number high molecular subtype, which are not classically associated with obesity. Despite this important molecular difference, the histology of all rat tumours in this study was similar to the histological appearances of obesity-related human endometrial cancer, furthering the idea that these rats have functional p53. The presence of functional p53 means this rat model is a more realistic model of human obesity-related EC. Our findings of MSI and high TMB suggest that this model is most closely related to the microsatellite unstable human molecular subtype. While diet did not significantly affect cancer incidence, it did alter the EC transcriptome and we identified 368 DEGs between tumours in the HFD group and tumours in the NCD group. We sequenced multiple samples from individual tumours to try to control for heterogeneity and were able to identify a number of clinically important alterations in human endometrial cancer. Specifically, we identified *Brd4* as a key upstream regulator in our dataset, which correlates with an extensive body of literature describing *Brd4* as a regulator of the oestrogen response genes in both endometrial and breast cancer [19,28]. 

Our diet-induced obesity study was used to identify two novel biomarkers for endometrial cancer, *A2M* and *CRABP2*, both of which appear to have important prognostic implications. *A2M*’s role in cancer biology is likely due to its ability to act as a signalling molecule and transporter of growth factors, such as TGFβ and IL6, and as a modulator of protease activity [29]. Serum *A2M* levels are inversely correlated with PSA in advanced prostate cancer, suggestive of a tumour-suppressive effect—as demonstrated in this study (Figure 3B) [30]. Serum levels of *A2M* decrease with age and obesity in humans [31]; our findings of a negative correlation between *A2M* mRNA expression and BMI in the TCGA cohort suggest that the BDII/Han rat model can be used to study specific interventions and identify molecular alterations that are relevant to human endometrial cancer.

We also present novel data on the role of *CRABP2* in endometrial cancer. *CRABP2* is a retinoic acid binding protein and along with the retinoic acid receptor (RAR) are important regulators of cellular turnover in the endometrium [32]. *CRABP2* is a highly conserved protein that functions in the retinoic acid signalling pathway, where it transports RA to the retinoic acid nuclear receptors (RAR and RXR), which regulates transcription of target genes involved in cell proliferation, differentiation and apoptosis. It also functions as a transcriptional coactivator: in the presence of *CRABP2*, transactivation by the RA–RXR–RAR complex is increased [33]. More recently, it has been reported that *CRABP2* can affect biological behaviour independently of RA or RAR [34].

Abnormal *CRABP2* expression is associated with numerous cancers, including non-small cell lung cancer (NSCLC) [35], pancreatic cancer [36] and serous ovarian cancer [37]. *CRABP2* ′s role in breast cancer may be related to oestrogen signalling that suppresses the ubiquitination of Lats1 to activate the Hippo pathway, which inhibits invasion and metastasis in ER-positive tumours but not in ER-negative tumours [38]. Given our recent description of the different roles of ER in breast and endometrial cancer, the interaction between *CRABP2* and ER in endometrial cancer warrants further investigation [39]. Our finding that *CRABP2* is an independent prognostic marker in endometrial cancer requires further validation; however, it is maybe an important novel biomarker in CN-high cancers, which have a poor prognosis.

## 4. Materials and Methods

Nineteen female, virgin BDII/Han rats were sourced from the Central Animal Facility (CAF) at the Hanover Medical School in Germany. The BDII/Han rats were divided into two groups: the normal chow diet group (NCD group, *n* = 7) and the high fat diet group (HFD group, *n* = 12). The rats were received from the CAF in Hanover at three months of age having been on normal chow diet from the time of weaning. Rats were age-matched and housed in spacious ventilated cages with hard floors in groups of 3–4 rats. They had 24 h access to ad lib food and water. The NCD group remained on normal chow diet until the study end at fifteen months of age. Normal chow with 3% kCal fat content was used as the chosen diet in the NCD group as it is generally accepted as the standard diet for research rats. The HFD group were immediately placed on a high fat diet from the time of arrival at three months of age and continued on this diet until the study end at fifteen months of age. The high fat diet included 60% kCal fat, 20% kCal carbohydrate and 20% kCal protein (Research Diets, NJ, USA). At fifteen months of age, CT imaging was carried out followed by euthanasia and systematic harvesting of animal organs was performed.

### 4.1. CT Imaging

CT scans were performed on a LabPET4 Triumph PET CT scanner (Trifoil Imaging Inc., Chatsworth, CA, USA) under isoflurane anaesthesia. Cone beam CT images (focal spot size, 33 mm) were acquired over ~ 5 min with 512 projections at a geometric magnification equal to a 1.3. X-ray tube voltage and the currents were 50 kVP and 350 µA, respectively, and the exposure was 232 ms. CT images were reconstructed with X-O CT software (V5.2.0.0 Trifoil Imaging Inc., Chatsworth, CA, USA), using a filtered back-projection algorithm to a voxel size of 0.2 × 0.2 × 0.2 mm^3^.

### 4.2. Assessment of Endometrial Cancer Burden

At the time of tissue harvest the animals were administered anaesthesia by inhalation of sevoflurane gas. A midline laparotomy and thoracotomy were performed. Euthanasia was performed by a terminal bleed from the aorta. Clinical photographs were taken and an assessment of the intrabdominal and peri-visceral fat was made and recorded on each animal. Complete organ harvest followed with organs dissected in order of research priority; uterus, ovaries, liver, omentum and lung. This allowed for complete staging of any endometrial cancer found.

### 4.3. Retrieval of Endometrial Tissue for Histology and RNA Isolation

A standardised method for the harvesting of uterine tissue was performed, as described by Ruehl-Fehlert [40,41]. One half of each tumour specimen was placed immediately in formalin and prepared for immunohistochemistry and histological assessment while the other half was “snap-frozen” in liquid nitrogen for interval RNA isolation and subsequent RNA sequencing. Parallel samples of normal endometrial tissue were also taken for histological and molecular assessment. Samples for histology were fixed in 4% formalin for 24 h before paraffin embedding. A total of 14 tumour samples were analysed from 4 affected rats in the NCD group while 12 tumour samples were analysed from the 4 affected rats in the HFD group. Histopathological assessment was performed by a histopathologist specializing in gynaecological oncology.

### 4.4. RNA Extraction, Library Prep and Sequencing

RNA was isolated from normal (*n* = 34 samples from *n* = 23 individual rats) and endometrial tumour (*n* = 31 from *n* = 13 individual rats) tissue using the EZ-RNA Total RNA Isolation Kit, according to the manufacturer’s protocol (Omega Bio-Tek, Norcross, GA, USA). Next-generation sequencing was performed by BGI Genomics (Shenzhen, China). After quality control procedures, individual RNAseq libraries were pooled based on their respective sample-specific, 7 bp adapters, and sequenced at a 2 × 75 bp paired-end read length using an Illumina NextSeq 500 sequencer. (see Appendix B).

### 4.5. RNAseq Processing, Differential Expression Analysis and Variant Calling

Raw sequence reads were first checked for quality using fastq software [42] and were then trimmed of low-quality reads using BBDuk [42]. The Kallisto [43] pseudo-aligner was used to map reads to the *Rattus norvegicus* Rnor_6.0 transcriptome from Ensembl release 94. The Sleuth [44] R package was used to read the Kallisto output, and to normalise and scale the transcript and gene expression values using the default settings. Limma [23], which uses a negative binomial distribution model to account for both biological and technical variation, was applied to identify statistically significant (corrected *p* value < 0.05) differentially expressed genes (DEGs), using the “voom” function and the robust empirical Bayes model. Limma was chosen as it allows correction for multiple samples taken from the same individual animal using the “duplicate Correlation” function. Variant calling was conducted using the HaplotypeCaller module of the Genome Analysis Toolkit following best practices for RNA. Filtering was conducted using custom Perl scripts. All RNAseq analysis methods are available from https://github.com/brucemoran/rdl_RNAseq.

### 4.6. Pathway and Functional Enrichment Analysis

Over-represented biological functions from the DEG sets were identified by functional enrichment analysis using Ingenuity Pathway Analysis (IPA; v. 8.8, Ingenuity Systems, Mountain View, CA, USA; http://www.ingenuity.com). The enrichment analysis was applied to the statistically significant differentially expressed genes.

### 4.7. TCGA Comparative Model

In order to draw comparisons between the animal data and human data we created a comparative data set from TCGA (*n* = 373) [7]. Two samples were excluded as raw sequence data was unavailable, leaving a final *n* = 371 samples. Further comparisons between the genes of interest from our animal model and human data were performed using Oncolnc [24], the Human Protein Atlas [25] and cBioportal [26]. Differences in the distribution of clinical data and tumour characteristics between genes of interest were evaluated using the chi-square test, Fisher exact test and independent t-test. Kaplan–Meier analysis and the log-rank test were used to illustrate differences between overall survival based on expression levels of *A2M* and Crabp2. Unsupervised decision tree analysis with a 10-fold cross-validation was used to identify the Z-score thresholds for the Kaplan–Meier analysis. Cox regression proportional hazards models were used to estimate the impact of the *A2M* and Crabp2 expression overall survival (OS) in both univariate and multivariate analysis. Variables found to be significant in the univariate analysis were included in the multivariate models. Analysis were performed using SPSS version 20.0 (IBM Corp, Armonk, NY, USA). All statistical tests were two-sided and *p* < 0.05 was considered statistically significant.

### 4.8. Ethics

This study was granted full ethical approval by the both the Animal Research Ethics Committee (AREC17-06) and the Health Product Regulatory Authority (AE1892/P119) on the 26th of July 2017.

## 5. Conclusions

Diet-induced obesity had a significant effect on the tumour transcriptome in a BDII/Han rat model of endometrial cancer. The BDII/Han rat model is an acceptable model to study obesity-related carcinogenesis and the impact of weight loss intervention on endometrial cancer biology and may be used to identify the clinically relevant genes of human endometrial cancer.

## Figures and Tables

**Figure 1 life-10-00188-f001:**
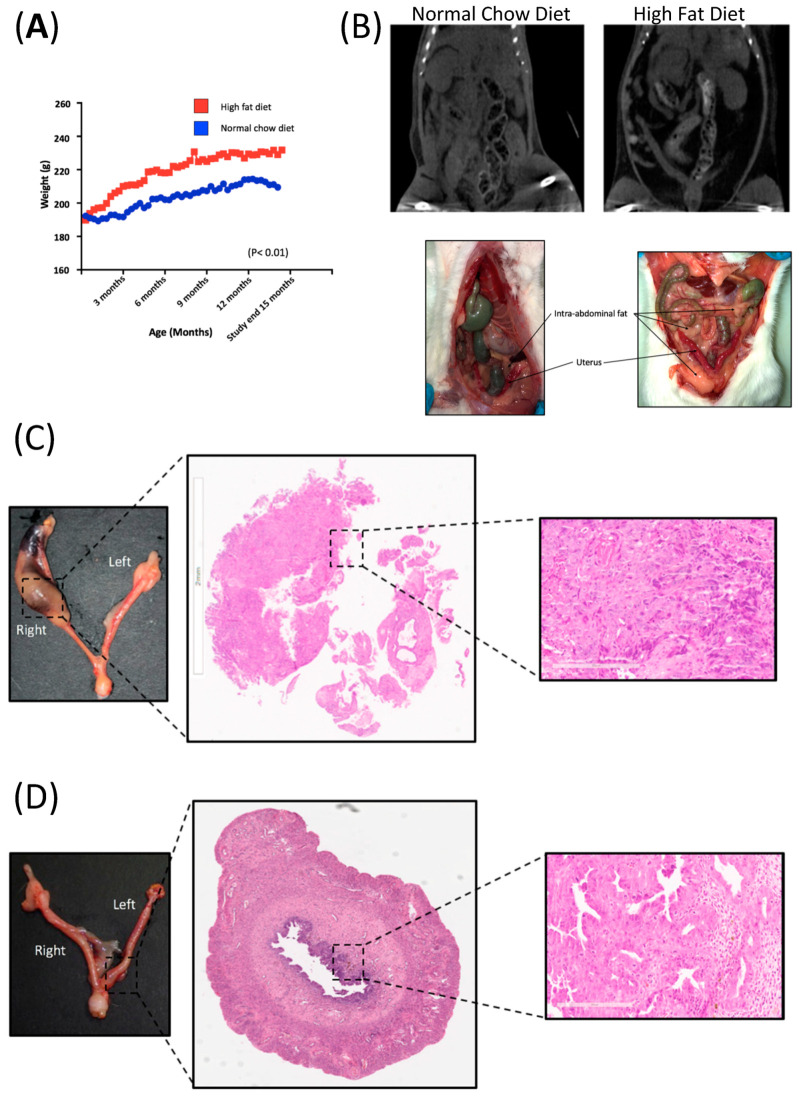
Diet-induced obesity in BDII/Han rats. (**A**) Graphical representation of the weight trajectory of the normal chow diet (NCD; *n* = 7, blue line) and high fat diet group (HFD; *n* = 12, red line). (**B**) Comparative CT imaging and photos taken at necropsy, showing the increase in volume of intra-abdominal fat in the HFD group (right) relative to rats from the NCD group (left). (**C**) Hysterectomy specimen from a BDII/Han rat from the high fat diet group. H&E showing endometrial adenocarcinoma in the right proximal uterine horn with a magnified (20×) view of the histology slide showing a solid area of the endometrial adenocarcinoma. (**D**) Hysterectomy specimen from a normal chow-fed BDII/Han rat. H&E showing endometrial adenocarcinoma in the left distal uterine horn, with a magnified (20×) view of the histology slide showing a gland-rich endometrial adenocarcinoma.

**Figure 2 life-10-00188-f002:**
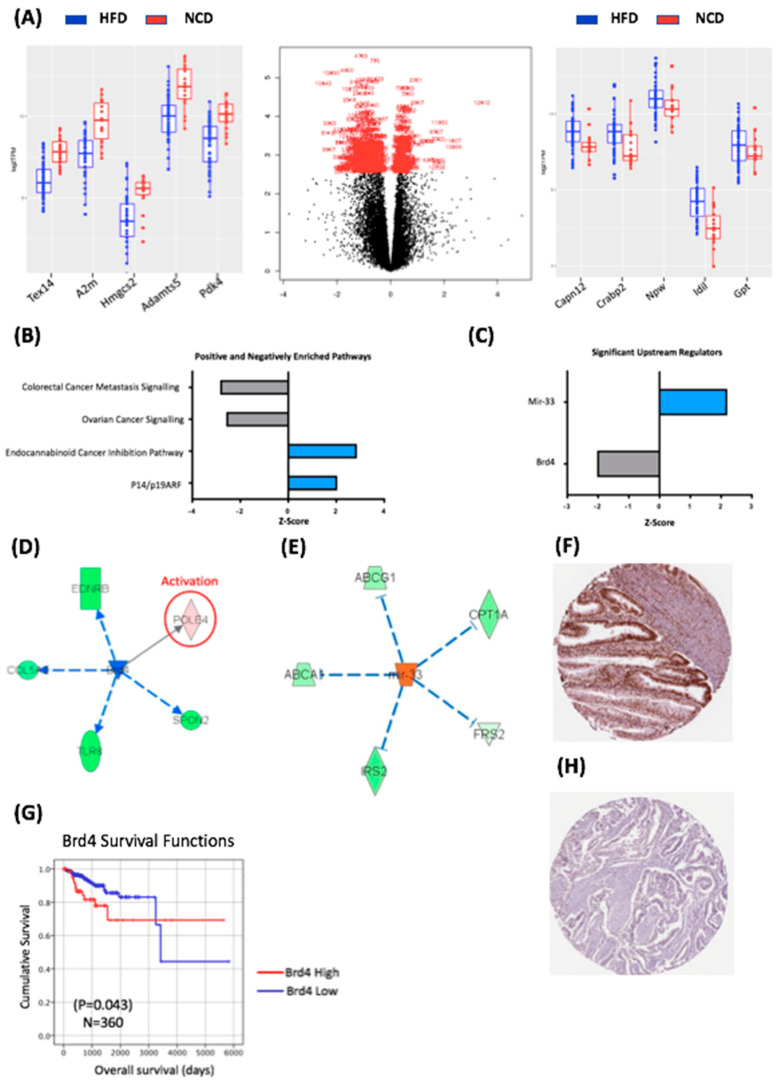
Transcriptomic characteristics of the impact of diet-induced obesity on BDII/Han endometrial tumours. (**A**) Volcano plot showing the differentially expressed genes between tumours from the HFD and NCD groups. The genes delineated in red are those that reached statistical significance (*p* < 0.05). The box plot on the left shows the top 5 genes with reduced expression in the HFD group relative to the NCD group. The box plot on the right shows the top 5 genes with increased expression in the HFD group relative to the NCD group. The HFD group is delineated in blue and the NCD group in red. (**B**) Top ranking canonical pathways in HFD and NCD tumour tissue. (**C**) Upstream regulators of DEGs in HFD and NCD tumours. (**D**) *Brd4* was a statistically significant upstream regulator, imparting a downstream effect on five DEGs. (**E**) *mir-33* was a statistically significant upstream regulator, imparting a downstream effect on five DEGs. (**G**) *Brd4* is prognostic in human endometrial cancer with decreased expression leading to lower overall survival. (**F**) Antibody stain showing high expression of Brd4 in 9/11 endometrial cancer samples from the Human Protein Atlas TCGA pathology database. (**H**) Corresponding antibody stain showing low expression of Brd4 in 3/11 human thyroid cancer samples from the Human Protein Atlas TCGA pathology database. The Human Protein Atlas was accessed July 2020.

**Figure 3 life-10-00188-f003:**
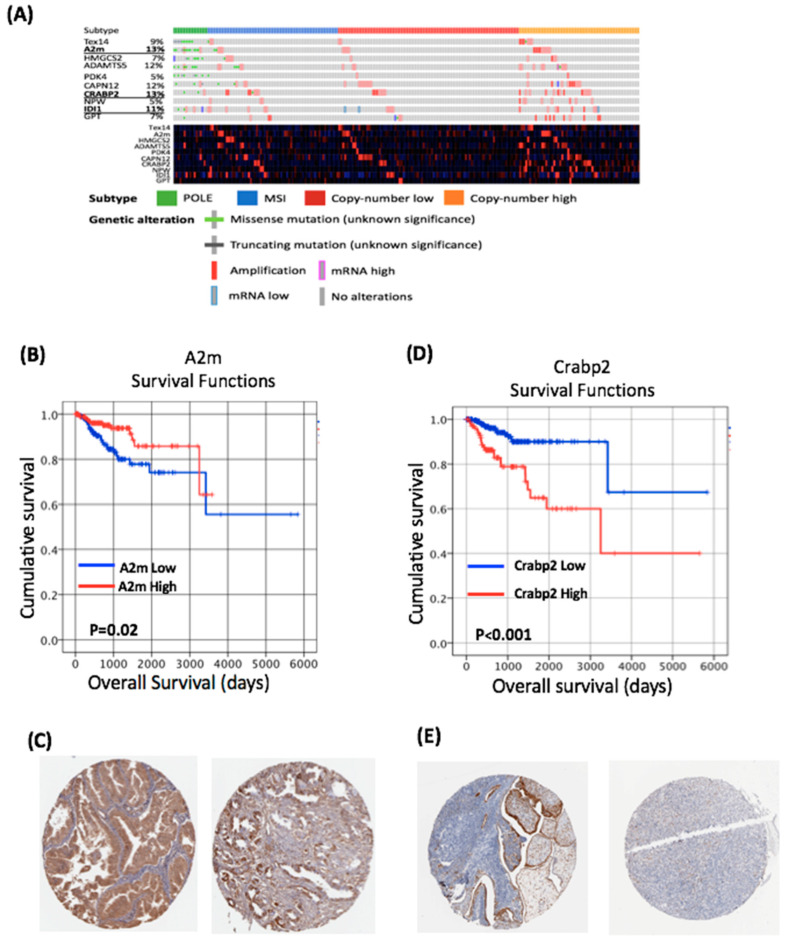
Genes altered by obesity in the BDII/Han model are clinically relevant in human endometrial cancer. (**A**) Genomic and transcriptomic alterations of the top 10 differentially expressed genes identified in human endometrial cancers. (**B**) Increased expression of *A2M* leads to a significant improved overall survival in human endometrial cancer. (**C**) Immunohistochemistry from the Human Protein Atlas demonstrates the expression of the *A2M* protein in human endometrial cancer. (**D**) Increased expression of *CRABP2* leads to a significantly improved reduced survival in human endometrial cancer. (**E**) Immunohistochemistry from the Human Protein Atlas demonstrates the expression of CRABP2 protein in human endometrial cancer. The Human Protein Atlas was accessed July 2020.

**Table 1 life-10-00188-t001:** *A2M* and Crabp2 mRNA expression and clinicopathological variables.

	*A2M* Low (*n* = 191)	*A2M* High (*n* = 180)	*p*-Value	Crabp2 Low (*n* = 265)	Crabp2 High (*n* = 106)	*p*-Value
Age Median (range)	63 (35–90)	63 (31–90)	0.665	63 (33–90)	65 (31–90)	0.093 *
BMI Median (range)	33.1 (19–82)	32.3 (17–61)	<0.001 *	33 (17–82)	31 (19–68)	0.485 *
Histology						
Endometrioid	147 (77)	158 (87)	0.003	238(90)	67 (63)	<0.001
Serous	32 (16.7)	21 (12)		20 (7)	33 (31)	
Mixed	12 (6.3)	1 (1)		7 (3)	6 (6)	
Stage						
1	120 (63.1)	132(74.2)	0.15	194 (74)	59 (55)	0.005
2	16 (8.4)	10 (5)		16 (6)	10 (9)	
3	44 (23.1)	30 (16.8)		47 (18)	28 (26)	
4	11 (5.4)	8 (4)		8 (2)	9 (9)	
Grade						
1	31 (16.2)	58 (32.2)	<0.001	74 (28)	15 (14)	0.001
2	50 (26.1)	55 (30.5)		77 (29)	28 (26)	
3	110 (57.7)	67 (37.3)		114 (43)	63 (59)	
TCGA molecular subtype						
POLE	11 (10.2)	6 (4.8)	0.28	16 (10)	1 (2)	<0.001
MSI	30 (28)	35 (28)		58 (35)	7 (10)	
CN Low	32 (30)	58 (46.4)		65 (39)	25 (37)	
CN High	34 (31.8)	26 (20.8)		26 (16)	34 (51)	

Values in parenthesis are percentages unless otherwise stated. Abbreviations: BMI—body mass index; SEM—standard error of the mean; TCGA—The Cancer Genome Atlas; MSI—microsatellite instable; CN—copy number. * Mann–Whitney U test.

**Table 2 life-10-00188-t002:** Cox regression analysis of OS for *A2M* and *Crabp2*.

		Univariate				Multivariate		
	**HR**	**95.0% CI**		***p*** **-Value**	**HR**	**95.0% CI**		***p*** **-Value**
**Overall Survival**		**Lower**	**Upper**			**Lower**	**Upper**	
*A2M* (low vs. high)	0.455	0.23	0.9	0.024	0.612	0.305	1.229	0.167
Age (continuous)	1.041	1.009	1.075	0.013	1.053	1.015	1.092	0.005
Histology (endometrioid vs. non-endometrioid	2.298	1.18	4.475	0.014	0.705	0.311	1.595	0.401
Stage (1 and 2 vs. 3 and 4)	4.654	2.448	8.849	<0.001	3.912	1.903	8.042	<0.001
Grade (1 and 2 vs. 3)	3.393	1.652	6.969	0.001	1.848	0.796	4.289	0.153
	**HR**	**95.0% CI**		***p*** **-Value**	**HR**	**95.0% CI**		***p*** **-Value**
**Overall Survival**		**Lower**	**Upper**			**Lower**	**Upper**	
Crabp2 (low vs. high)	3.554	1.875	6.735	<0.001	3.134	1.524	6.443	0.002
Age (continuous)	1.041	1.009	1.075	0.013	1.046	1.011	1.083	0.01
Histology (endometrioid vs. non-endometrioid)	2.298	1.18	4.475	0.014	0.451	0.193	1.052	0.065
Stage (1 and 2 vs. 3 and 4)	4.654	2.448	8.849	<0.001	3.875	1.868	8.036	<0.001
Grade (1 and 2 vs. 3)	3.393	1.652	6.969	0.001	1.802	0.775	4.19	0.171

Abbreviations: A2M—alpha2macroglobulin; CI—confidence interval; HR—hazard ratio; OS—overall survival.

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
