# Peer review of "The Molecular Effects of a High Fat Diet on Endometrial Tumour Biology"

_life, 2020, doi:10.3390/life10090188_

Round 1

Reviewer 1 Report

I read with great interest the work entitled "The molecular effects of high fat diet on endometrial tumour biology".

The Authors conducted their studies using an animal model (BDII/Han rat) and datasets containing information on human endometrial cancer. They indicated that the mentioned animal model is suitable for the study of diet induced obesity in EC, and due to the molecular and histological similarity between this model and the human obesity-related EC, it is possible to identify useful biomarkers in human EC.

Comments:

1. Line 46 - It is worth mentioning whether "the most common cancer" refers to the US or worldwide.

2. The study material consisted of samples from 19 rats (7 NCD, 12 HFD). How was sample size calculated?

3. Is there a reason for the unequal sample sizes? The difference of 5 rats does not seem great, however, considering the small size of the groups, the HFD group is almost 2 times larger than the NCD group.

In addition, EC was detected in 4 rats from the NCD group and 4 from the HFD group, and more samples (replicates) from the HFD group were used in the further stages of the analysis (e.g. RNAseq, 18 NCD vs. 47 HFD)

4. When using online databases/tools, it is good to give information when they were accessed.

5. Lines 295-297 - The sentence "A post-hoc power analysis demonstrated a study to demonstrate a statistically significant difference in endometrial cancer incidence in the NCD and HFD groups would require 116 rats." could be changed to:

"A post-hoc power analysis showed that the study would require 116 rats to demonstrate a statistically significant difference in the endometrial cancer incidence in the NCD and HFD groups."

6. Lines 367-368 - I think adding a verb to this sentence will improve the flow, e.g. "The high fat diet included 60% kCal fat....."

Or, you can leave it as it is, but enclose it in parentheses.

7. I recommend re-reading the text and checking it for typos and editing errors, e.g.:

Line 28 - Two periods at the end of the sentence.

Line 29 - Versus is written as Vs but it's v's in the rest of the work.

Line 55 - adenocarcinoma's or adenocarcinomas?

Line 92 - Check the position of the parentheses.

Line 222 - removed instead of remvoed.

Line 381 - There is a period at the end of the section name.

Line 399 - Kit's company is provided but city and country are missing.

Line 437 - The country is missing.

Author Response

  1. Line 46 - It is worth mentioning whether "the most common cancer" refers to the US or worldwide.

 This has been changed to ‘worldwide’.

  1. The study material consisted of samples from 19 rats (7 NCD, 12 HFD). How was sample size calculated?

These numbers were chosen on the presumption that as many as 90% of the animals used would develop endometrial cancer by the age of 13 months. A larger number of animals were chosen for the HFD group as it was anticipated that this group may experience and earlier mortality because of the calorie rich diet.  

  1. Is there a reason for the unequal sample sizes? The difference of 5 rats does not seem great, however, considering the small size of the groups, the HFD group is almost 2 times larger than the NCD group. In addition, EC was detected in 4 rats from the NCD group and 4 from the HFD group, and more samples (replicates) from the HFD group were used in the further stages of the analysis (e.g. RNAseq, 18 NCD vs. 47 HFD).

This is something we have explained more clearly with a more detailed breakdown of normal and tumour samples from each group detailed in lines 109-111. We have also added an additional supplementary table S5 that details the number of samples taken from each group of rats. Please note, as now stated in lines 109-111, there were 14 tumour samples analysed from the NCD group and 10 tumour samples from the HFD group. 

  1. When using online databases/tools, it is good to give information when they were accessed.

This has been addressed in the text where present. Please see track changes. 

  1. Lines 295-297 - The sentence "A post-hoc power analysis demonstrated a study to demonstrate a statistically significant difference in endometrial cancer incidence in the NCD and HFD groups would require 116 rats." could be changed to:

"A post-hoc power analysis showed that the study would require 116 rats to demonstrate a statistically significant difference in the endometrial cancer incidence in the NCD and HFD groups."

This has been changed in the text to the suggested sentence. 

  1. Lines 367-368 - I think adding a verb to this sentence will improve the flow, e.g. "The high fat diet included 60% kCal fat....." Or, you can leave it as it is, but enclose it in parentheses

This has been changed in the text.

  1. I recommend re-reading the text and checking it for typos and editing errors, e.g.:

Line 28 - Two periods at the end of the sentence. Corrected in text.

Line 29 - Versus is written as Vs but it's v's in the rest of the work. Corrected in text.

Line 55 - adenocarcinoma's or adenocarcinomas? Corrected in text.

Line 92 - Check the position of the parentheses. Corrected in text.

Line 222 - removed instead of remvoed. Corrected in text.

Line 381 - There is a period at the end of the section name. Corrected in text.

Line 399 - Kit's company is provided but city and country are missing. Corrected in text.

Line 437 - The country is missing. Corrected in text.

Reviewer 2 Report

The work is a very interesting scientific report. The animal model could benefit further research into endometrial cancer in obese women.
Understanding the prognostic factors can further contribute to a better, more personalized treatment. Correct work in terms of content and form.

Author Response

Many thanks to the reviewer for the kind feedback.